# NAM-NMM Temperature Downscaling Using Personal Weather Stations to Study Urban Heat Hazards

Martina Calovi [1,*], Weiming Hu [2], Guido Cervone [2] and Luca Delle Monache [3]

1    Department of Geography, Norwegian University of Science and Technology, 7491 Trondheim, Norway
2    Department of Geography, Institute for Computational and Data Sciences and Earth and Environmental Systems Institute, The Pennsylvania State University, University Park, PA 16801, USA; weiming@psu.edu (W.H.); guc18@psu.edu (G.C.)
3    Scripps Institution of Oceanography, University of California, La Jolla, CA 92093, USA; ldellemonache@ucsd.edu
*    Correspondence: martina.calovi@ntnu.no

**Abstract:** Rising temperatures worldwide pose an existential threat to people, properties, and the environment. Urban areas are particularly vulnerable to temperature increases due to the heat island effect, which amplifies local heating. Throughout the world, several megacities experience summer temperatures that stress human survival. Generating very high-resolution temperature forecasts is a fundamental problem to mitigate the effects of urban warming. This paper uses the Analog Ensemble technique to downscale existing temperature forecast from a low resolution to a much higher resolution using private weather stations. A new downscaling approach, based on the reuse of the Analog Ensemble (AnEn) indices, resulted by the combination of days and Forecast Lead Time (FLT)s, is proposed. Specifically, temperature forecasts from the NAM-NMM Numerical Weather Prediction model at 12 km are downscaled using 83 Private Weather Stations data over Manhattan, New York City, New York. Forecasts for 84 h are generated, hourly for the first 36 h, and every three hours thereafter. The results are dense forecasts that capture the spatial variability of ambient conditions. The uncertainty associated with using non-vetted data is addressed.

**Keywords:** ensemble modeling; temperature forecast; urban environments; volunteered geographic information; spatial downscaling; private weather station

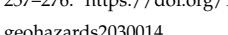



## 1. Introduction

Rapid temperature increases, exacerbated by climate change, can pose unacceptable risks to coastal areas and low-lying islands due to seawater rise, and to urban areas and megacities in particular, where the urban heat island effect amplifies this trend [1]. Being able to predict with confidence and accuracy temperatures in an urban environment is a challenging and necessary task to mitigate the uncertainty introduced by climate change.

*1.1. Urban Temperature Increase Due to Climate Change*

Climate change model simulations show average surface temperature increases between 2 °C and 6 °C by the end of the 21st century (https://earthobservatory.nasa.gov/features/GlobalWarming (accessed on June 2021)). This drastic increase suggests that in the near future short-term weather forecasting will need to address more frequent scenarios that are presently considered rare or extreme. This paper focuses on short-term weather forecasting, and it is motivated by the general trend of changing weather conditions that require forecasts with a very high spatial resolution in an urban environment.

Approximately 80% of the United States population lives in metropolitan areas, making these densely populated areas most at risk and the first to experience the effects of climate change (https://nca2014.globalchange.gov/report/sectors/urban (accessed on

June 2021)). Climate change hits urban environments by threatening all the basic infrastructures, from the built to the natural and social systems, especially due to the increased risk, frequency and intensity of extreme events. Heat waves, precipitations, and floods are among the most dangerous. Hazards aggravate the already precarious conditions of the most vulnerable population, who is proportionally more exposed and sensitive to the effects of climate change and lack in dynamic adaptive capacity. It was shown that social inequalities, connected especially to age, ethnicity, gender, income, health, and disability, increase differences in coping and adaptive capacities [2,3].

This particular work focuses on creating a baseline to study temperatures that is most useful to forecast heat waves at a higher resolution than regional forecast models are generally run. This method is paramount to forecast rare and extreme events in urban areas, and in turn to improve population urban vulnerability.

Weather and climate models have a spatial resolution in the order of kilometers, and are not capable of resolving temperature forecasts at an urban scale and capture the high temperature variability where buildings, parks, open areas, and other characteristics of the built environment can significantly cool or heat ambient temperature.

Temperature predictions in urban environments are challenging because of the high spatial gradient that exists between nearby areas. The sharp temperature changes are due to the characteristics of the built environment that pair hotter concrete and cement structures to cooler vegetation and shaded areas. These temperature changes generally occur at a meters spatial scale, which is much smaller than the kilometers spatial resolution of weather models. Models do not forecast temperature at such high resolution mainly due to their resolution and the specification of land surface type, and because of uncertainty in parametrization and in the computation of fluxes [4].

Furthermore, generating forecasts at an ultra-high resolution is very computationally expensive, and requires substantial investments in computing infrastructure. Improving current forecasts in terms of accuracy and spatial resolution is thus fundamental.

The proposed method was tested in the Manhattan Borough of New York, New York, which was chosen because of the very high population density, paired with a large number of available weather stations. Current estimates give a population of more than 1,600,000 inhabitants in the Manhattan Borough of New York (https://www.census.gov/quickfacts/newyorkcountymanhattanboroughnewyork (accessed on January 2021)). The United States Environmental Protection Agency (EPA) suggests that annually the average temperature of a city counting 1 million or more can be 1–3 °C warmer than the surrounding areas, and with up to a 12 °C difference during the evening hours (https://www.epa.gov/heatislands (accessed on January 2021)).

This effect is called Urban Heat Island (UHI) and it refers to city centers that have higher temperatures than nearby areas. Susca et al. [5] found a 2.5 °C higher average temperature between New York City and the nearby areas. Rizwan et al. [6] defined the UHI as one of the most serious problems of the 21st century for humans and it is increasing due to the growing urbanization and industrialization. Increased air temperature is caused by a greater heat absorption by urban materials when compared to vegetation. These urban materials retain a high temperature during nighttime by releasing heat slowly. Furthermore, tall buildings limit air flow, and reduce the number of green areas that help the natural convective cooling [7] and the radiative effects [8,9]. Especially during extreme temperature events, the UHI effect has the potential to maintain nighttime temperatures at a level that affects human health and comfort.

An important aspect that characterizes forecasts is their spatial scale. To mitigate risk and develop effective strategies to cope with heat events, decision makers—in a variety of socio-economic sectors—need information at the scales for which decisions are made [10–12]. Problems that affect a population at a local scale, require forecasts at a much finer resolution provided by weather models [10,12]. To provide actionable forecasts to decision makers, given the limitations discussed previously of running models at ultra-high resolutions, it is necessary to develop effective downscaling processes [12].

*1.2. Temperature Downscaling in Urban Areas*

There are two main approaches for downscaling: dynamical and statistical. Dynamical downscaling employs regional climate models and forces them to use global climate model boundary conditions. It is a computationally intensive process, and it requires changing the parametrization of the model to generate different outcomes. Statistical downscaling, instead, is implemented by developing empirical relationships between historical and current forecasts. High-resolution probabilistic forecasts are made by altering a single deterministic model prediction with historical measurements [11,13]. According to Trzaska and Schnarr [11] an effective statistical downscaling requires a strong relationship between the starting large-scale model and the finer output local scale.

The first application of a downscaling process using the AnEn technique is presented in Keller et al., 2017. They successfully tested a method to statistically downscale precipitation estimates from a regional reanalysis for Germany, and they generated synthetic probabilistic observations for periods where measurements were not available [14].

In this study, the AnEn technique is used to downscale temperature forecasts in an urban environment using data at different spatial and temporal resolutions. A novel downscaling indices method, based on the AnEn technique, is introduced and presented with this study.

To pursue this research intent, non-authoritative data sources have been used as past repositories of observation measurements. These data are called Volunteered Geographic Information (VGI), and their main characteristic is that they are voluntarily contributed by the public [15]. Often used in assessing emergency situations, these data are collected and distributed outside of traditional, authoritative emergency management methods and agencies [16], and often contain precious temporal and spatial information [15,17,18]. These data provide a large, rapidly changing, dynamic dataset that not only complements authoritative observations, but can also add measurements that better characterize specific phenomena [19]. Weather Underground (WU) stations data are used as a class of VGI. WU is a free-to-use network that provides a platform to voluntarily upload data collected through Private Weather Station (PWS)s. WU data are considered VGI because they are freely contributed by the community. Many weather data are collected using PWSs, and can be downloaded free of charge using a provided Application Programming Interface (API) from the WU website. Although WU data are global in nature, their largest coverage is found in the USA, Europe, Japan and Australia. There are over 250,000 PWSs registered in the WU network, and their distribution follows population density, thus most of the stations are in urban areas.

The dense number of network nodes in urban areas allows identification of a general temperature trend, which is usually different but within a threshold of the Aviation Weather Center—METeorological Aerodrome Reports (METARs) stations, but at the same time, capture small temperature variations which are shown by the PWSs that are spatially distributed over the study area. An important contribution of this research consists of controlling the quality of these measurements, characterizing their reliability, and testing their use in high-resolution forecasting.

The paper is organized as follows. Section 2 describes the materials and the method: it presents the data, the WU data quality controls, and it describes the AnEn technique and the applied downscaling technique. Section 3 presents the results of the spatial downscaling and the verification measurements used to control the performance. In Section 4 the results are discussed while drawing the conclusions.

## 2. Materials and Methods

The complex nature of weather prediction at high spatial resolution, challenged the authors in integrating multiple datasets that differ by their nature. The fusion of such diverse datasets has been addressed through the AnEn technique [20] that allows for the generation of a Probability Distribution Function (PDF) of expected outcomes from a current deterministic forecast and corresponding sets of historical forecasts and verifying

observations, without expecting the input data to be from the same model [20]. In this project a deterministic forecast has been addressed, where a deterministic forecast consists of a single value for each time in the future for temperature [21].

*2.1. Data Sources*

The entire analysis has been conducted over two years of data, 2015 and 2016, focusing on the area of Manhattan, New York.

Historical forecasts. For this study, the North American Mesoscale Forecast System (NAM-NMM) model is used as the initial deterministic atmospheric forecast. The North American Mesoscale Forecast System (NAM-NMM) model is one of the major weather models used to produce forecasts, and is run by the National Centers for Environmental Prediction (NCEP). The model has four cycles per day, running at 0000, 0600, 1200 and 1800 UTC, covering a whole time window of 84 h [22]. Each cycle represents a FLT, defined as the time that elapses between the forecast and the event that has been forecasted. For the first 36 h, the FLTs are hourly, thereafter they became every three hours up to covering the 48 remaining hours. The model has a grid horizontal resolution of 12 km, over the Continental US. The NAM-NMM simulates about 400 weather variables from 60 vertical layers. In this study, 5 variables at their corresponding vertical layers have been used as predictors to forecast the temperature. Each of these 5 variables has been assigned a weight (see Table 1), based on the weight optimization experiments carried out by [23–25].

The NAM-NMM data have been preprocessed using the supercomputer Cheyenne from the National Center for Atmospheric Research (NCAR), given the huge original size of the model dataset.

**Table 1.** NAM-NMM predictor variables used in the study.

| Parameter | Abbreviation | Vertical Level | Weight |
|-----------|--------------|----------------|--------|
| wind speed | WSPD | 10 mASL | 0.1 |
| wind direction | WDIR | 10 mASL | 0.1 |
| Temperature | TMP | Surface | 0.4 |
| Relative humidity | RH | 10 mASL | 0.4 |

Historical observations. The observation data used in this study have been downloaded from the WU website—access and data download date back to January 2017. The data refer to a time window that spans from 1 January 2015, 00:00 to 31 December 2016, 23:59; an initial total of 110 PWSs have been identified covering the Manhattan borough. The observation measurements collected through the WU stations are available at different time resolutions; for the purpose of this study, they have been interpolated to the hourly FLTs of the forecast model.

*2.2. Weather Underground Data Quality Control*

Using WU data to generate reliable temperature forecasts presented several challenges. First, their spatial distribution is not uniform, it varies from a dense network in urban areas and sparse coverage in more rural areas. However, within urban areas, there can be significant differences in spatial coverage. Generally, residential areas with higher median income tend to have denser coverage. Second, data are collected asynchronously, and each PWSs can update information at a different time and with different temporal resolutions. For example, some stations update data every few seconds, while others only a few times throughout the day.

The temporal update usually depends on the network connection available to the station, with higher update frequency when faster and reliable networks are available (https://www.wunderground.com/about/data (accessed on May 2019)). Some limitations are in the preprocessing and analysis of WU data which are time consuming but are essential to ensure the robustness and reliability of their use in the proposed framework [26].

A quality control was run on all the 110 stations available over the period and area of this study.

1. All the stations with no data were excluded, reducing immediately the stations number to 105.
2. Among the remaining stations, four are part of the METARs weather station network, namely: KLGA—LaGuardia Airport, KTEB—Teterboro Airport, KJRB: Wall Street station and KNYC—Central Park. Stations belonging to the METARs network have a more rigorous quality control, as they are officially operated by entities and not private citizens. This does not mean that do not include errors or noise, but rather that they can be assumed to be more reliable than regular PWSs. Because of this consideration, these four stations were used as reliable representatives for the area, and used to filter PWSs that drastically deviated from their measurements. Going forward, we assume to have 105 valid stations, four of them being the more reliable METARs stations, and 101 regular PWSs.
3. A set of monthly statistics for all the 105 WU stations were calculated, including average, minimum, first quartile, median, third quartile, and maximum.
4. Additionally, statistics were computed between the four METARs stations and all the 101 PWSs.
    a   The monthly Root Mean Square Error (RMSE). This shows the monthly variation of each of the PWS with respect to METARs stations and shows bias that might change with values (e.g., inability to identify extreme values), season (for example installation problems related to shading). This method was not used to eliminate individual stations, but individual hourly observations that deviate from the general trend. This step is further explained later.
    b   The yearly RMSE. This shows the yearly variation among all values, and is primarily an indication of very strong deviation from the METARs representative. Examples are stations that are installed indoors or close to exhaust vents, recording values that are not usable for the intended purposes of this article. This method was used to eliminate 17 stations from the analysis. This step is further explained later.
5. Five more PWSs show drastically different values than the four METARs stations; however, they show very strong similarities among each other, creating a nice cluster. They were separated and further investigated and analyzed. Their anomaly distributions make them outliers stations compared to the distribution of the other PWSs; however, their similarities, which do not correspond to spatial proximity, make them an interesting case for future studies. However, they were excluded from the analysis in this work.

In total, 27 (17 + 5 + 5) PWSs, out of the 110, have been excluded from the analysis, because they did not pass the quality controls, while 83 stations (including the 4 METARs stations) did and have been included in the study.

2.2.1. Filtering of Entire Stations Based on Yearly Statistics

Figure 1 shows a qq-plot of the yearly RMSE statistics between the valid stations. The dotted red line represents the maximum threshold (3 °C) value from which the average of the 101 WU stations can deviate from the mean temperature of the METARs stations to be included, otherwise excluded. The green points represent the included WU stations, while the reds the excluded WU stations (because they showed a mean annual temperature that deviates more than 3 °C from the threshold). According to the threshold shown in this figure, 17 stations were eliminated (red dots) as also explained in 4b.

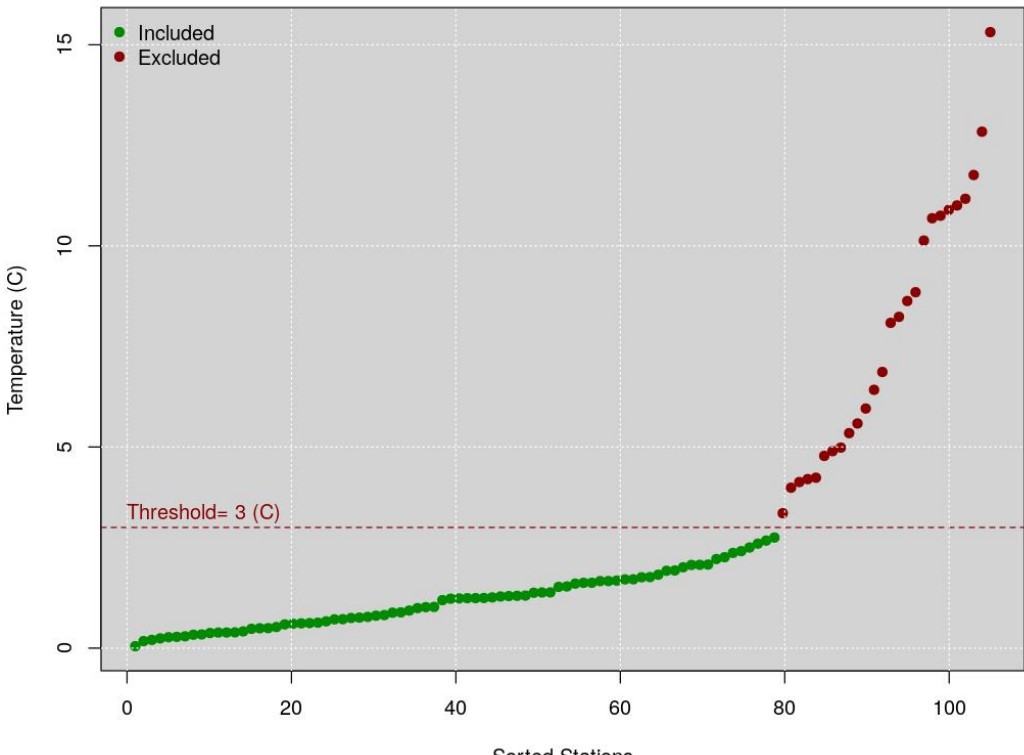

**Figure 1.** WU data quality control—QQ graph of the average annual temperature of the PWSs. The dotted red line represents the maximum threshold (3 °C), value from which the average of a PWS can deviate from the mean temperature of the stations to be included, otherwise it will be excluded.

### 2.2.2. Filtering of Entire Stations Based on Monthly Statistics

Figure 2 shows the hourly mean temperature of each PWS. In both the graphs, the black lines represent the mean temperatures of the PWSs, while the dashed yellow line represents the mean temperature of the four METARs stations. The upper graph shows the WU stations represented with the green points in Figure 1 (their mean's deviation from the mean temperature of the officials METARs stations is within the threshold); while the lower graph shows the WU stations represented with the red points in Figure 1 (their mean's deviation from the mean temperature of the officials METARs stations is higher than the threshold).

To note that each of the 101 PWSs monthly mean temperatures has been compared individually with the mean temperature of the four METARs stations, to check their average deviation from the threshold validity.

### 2.2.3. Filtering of Individual Hourly Values Based on Monthly Statistics

Figure 3 shows, as an example, the monthly temperature of station KNYMANHA7. The black line represents the monthly mean temperature of the METARs stations. The red line represents the monthly mean temperature of station KNYMANHA7. The grey line shows the daily temperature of station KNYMANHA7. The time steps correspond to each month. The red filled points represent the time steps included in the analysis (their mean's deviation from the mean of the officials METARs stations is within the 3 °C threshold); the empty red circles represent the time steps excluded, because their mean's deviation from the mean of the officials is higher than the threshold.

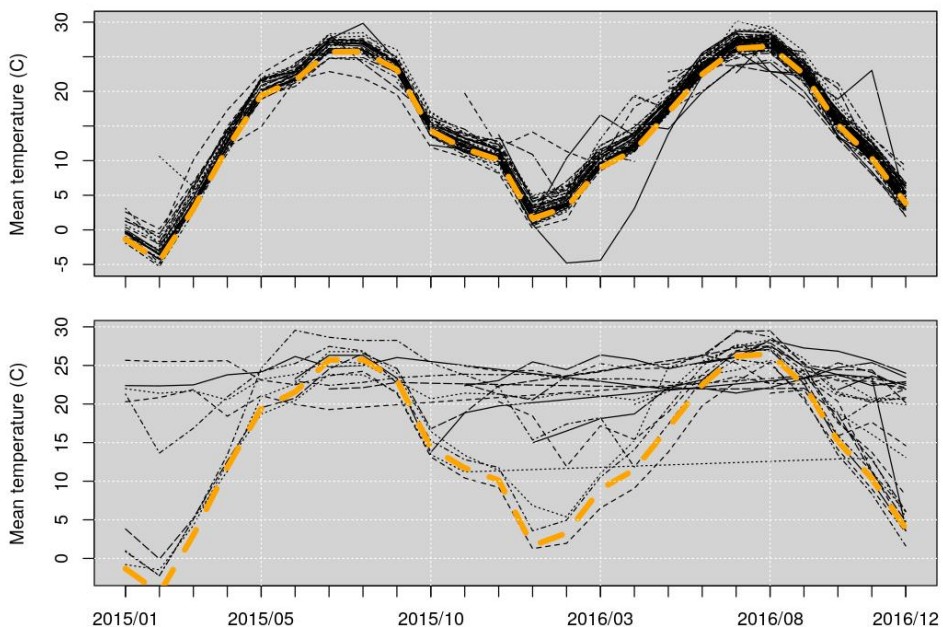

**Figure 2.** WU data quality control—Mean temperature of PWSs. In both the graphs, the black lines represent the mean temperatures of the PWSs, while the dashed yellow line represents the mean temperature of the four METARs stations. The upper graph shows the stations represented with the green points in Figure 1 (mean within the threshold); while the lower graph shows the stations represented with the red points in Figure 1 (mean higher than the threshold).

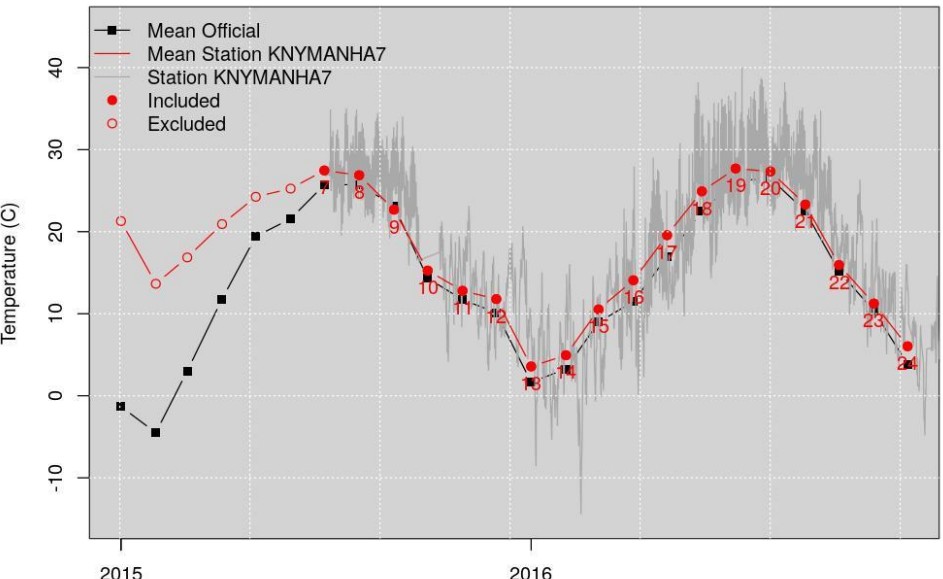

**Figure 3.** WU data quality control—Monthly temperature of station KNYMANHA7. The black line represents the monthly mean temperature of the stations. The red line represents the monthly mean temperature of station KNYMANHA7. The grey line shows the daily temperature of station KNYMANHA7. The time steps correspond to each month. The red filled points represent the time steps included in the analysis (the monthly mean temperature deviation is within the 3 °C threshold); the red circles represent the time steps excluded, because their monthly mean temperature deviates more than the threshold.

The result of this filtering of stations and individual values, provides a dataset that, while being contributed and not vetted, agrees with the overall trend shown by the METARs stations.

### 2.3. NAM-NMM Station Identification

The NAM-NMM is initialized four times a day at 0000, 0600, 1200, and 1800 UTC and consists of different components, where the main one is the model simulation that cover the CONUS 12 km parent domain, for 84 h into the future [27,28]. A filtering process, based on the Euclidean distance, has been used to select the closest NAM-NMM grid point to each WU station. As a result of this process, three NAM-NMM grid points have been identified in the study area. In Figure 4 the three red circles represent these NAM-NMM grid points.

For each of the three grid points, the AnEn technique was individually run to generate the analogs, and their RMSE was calculated and used as evaluation measurement to decide which one of the three grid points performs better. The upper-east grid point shows a mean RMSE of 2.017 °C, the lower-west point shows a mean RMSE of 2.04 °C, while the upper-west point shows an average error equal to 1.999 °C. The upper-west grid point, the one that showed a lower RMSE, has been selected and used in the analysis as the historical forecast repository.

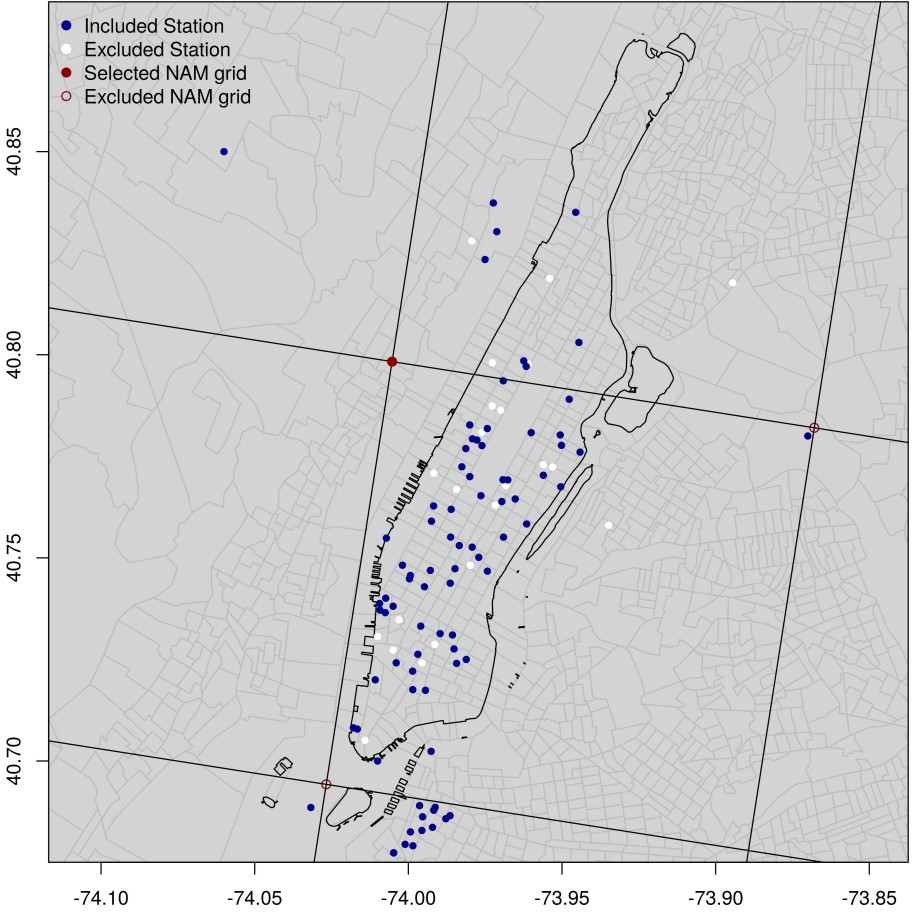

**Figure 4.** Stations spatial distribution—The blue points represent the PWSs that passed the quality controls and have been included in the analysis; while the white points represent the stations that have been excluded. The three red circles represent the NAM-NMM grid points. The red one is the one selected as a past forecasts repository.

Moreover, following the main objective of the study to investigate, predict and downscale temperature at the urban environment microscale, the upper-west grid point, represents the location that is more realistic and consistent with the study setting environment. The upper-east grid point is located close to the LaGuardia Airport and the lower-west represents a location over the water, see Figure 4.

*2.4. Analog Ensemble*

The AnEn technique generates probabilistic predictions using a single deterministic Numerical Weather Prediction (NWP), a set of past forecast predictions, and their corresponding observations [20].

AnEn technique compensates for the intrinsic biases that, as often happens, ensemble models carry with them, calibrating the forecasts using real-world observations [29]. There are many methodologies to perform this calibration (e.g., Model Output Statistics (MOS)), and they have shown to drastically improve the accuracy and uncertainty of the forecasts. The main difference using an analog-based technique such as AnEn, is that the analogs that make up the ensemble forecasts are generated directly using the observations, the same ones that are used in MOS-based post-processing to calibrate the output of the model. Therefore, AnEn forecasts do not need additional post-processing calibration because they are generated using the real-world observations, thus being intrinsically calibrated and bias corrected.

The main assumption is that if an error exists between a current forecast and a set of observed values, a similar error occurred between past similar forecasts and their corresponding observed values [20,21]. Therefore, the AnEn method is best suited when current and historical forecasts are generated using a static model that did not undergo any updates over the data time window used as historical forecasts. AnEn normally suffers from model updates that alter the expected error between forecasts and observations. More specifically, the AnEn technique assumes that the relationship between model forecasts and observations is constant, or that the relationship can be predicted, considering that weather patterns are associated with similar model errors that will be corrected using the AnEn forecasts and the corresponding observations.

Additionally, the AnEn technique can capture flow-dependent error characteristics and shows superior skills in predicting rare events when compared to state-of-the-art post-processing methods [20,29]. In addition to these engaging features, the AnEn technique is capable of computational scalability and high-resolution forecasts, and it does not rely on initial conditions, model perturbation, or post-processing requirements [30]. A reliable AnEn needs at least several months of past observations and predictions data that span the seasons relevant to the prediction of interest. The AnEn technique has been already successfully implemented in short-term prediction of 10 m and 80 m wind speed and 2 m temperature [23] and in the prediction of wind and solar power [21,30–33] and renewable energy [34,35]. Djalalova in 2015 applied the methodology to a study of the air quality [13,14,36] used it to study dynamical and statistical downscaling.

For each FLT, the AnEn technique creates an ensemble of corresponding sets of observations from the historical dataset [20,21]. These measurements are those concurrent with past forecasts at the same FLT, chosen across the past forecasts most similar to the current forecast, and as a result, AnEn predictions have the benefit of being calibrated [35]. The AnEn technique can estimate the probability distribution of the observed future value of the predictand variable given a model prediction, which can be represented through the equation [35]:

$$F\left(y \mid x^f\right) \tag{1}$$

where at a given time and location, $y$ is the unknown observed future value of the predictand variable and $f$ the values of the predictors from the deterministic model prediction at the same location and over a time window centered at the same time [37].

The similarity metric that determines the best match, is given by the following equation of $F_t$ that represents the forecast that must be corrected at a given time *t* and at a specific location in space [20]:

$$\|F_t, A_t\| = \sum_{i=l}^{N_v} \frac{W_i}{\sigma i} \sqrt{\sum_{j=-\tilde{t}}^{\tilde{t}} (F_{i,t+j} - A_{i,t+j})^2} \qquad (2)$$

where $A_t$ is an analog forecast at time *t* before $F_t$ was issued and at the same location; $N_v$ and $W_i$ are the number of predictors and their weights, respectively; $\sigma_i$ is the standard deviation of the time series of past forecasts of a given variable at the same location; t is an integer equal to half the width of the time window over which the metric is computed; and $F_{t,t+j}$ and $A_{i,t'+j}$ are the values of the forecast and the analog in the time window for a given variable [20,29].

The similarity metric identifies the best match between a current deterministic multivariate prediction and past deterministic multivariate predictions, and allows comparison of the current deterministic prediction with each of the historical deterministic predictions.

The predictor weighting parameter ($W_i$) has been proved to have impacts on the performance of the AnEn technique. Ref. [23] investigated optimal weighting strategies for the AnEn at five specific wind farms for wind power forecasting. Ref. [30,37] used optimal weighting at three specific solar farms for solar energy forecasting and determined that the AnEn performed best when the predictors were optimally weighted. Ref. [25] in their new undergoing research project, are also investing the effects of the weighting parameter and its importance in the equation.

The algorithm chooses the best matching ensemble members, for one day and for a specific deterministic forecast, which in this study is represented by the parameter temperature, while the analog members refer to the past best matching days and are a synthesis of all the predictor's combinations. The AnEn technique chooses the past temperature measurements based on the multivariate L2 norm described in Equation (2).

The key features of the AnEn technique include [20,29,30,37]:

1. The generation of an ensemble without the need for any perturbation strategy (e.g., of the initial conditions, physical parameterizations, models, etc.);
2. The need in real time of only one deterministic prediction, which could result in significant real-time computational savings concerning traditional ensemble methods based on several model runs. Alternatively, if the same resources as the ones needed to generate a traditional ensemble are used, AnEn allows for the generation in real time of a higher fidelity prediction (with finer horizontal and vertical resolution);
3. The prediction is based on past observed values, i.e., not on model estimates, which results in low-bias and well-calibrated forecasts;
4. More accurate deterministic predictions of the deterministic system used to generate AnEn;
5. Sharp and reliable probabilistic predictions.

At a given time and location, the future value of the predictand variable is given by the ensemble of observed values that are associated with the most similar forecasts to the current deterministic forecast [20]. Each forecast is composed of a multivariate vector of geophysical variables, whereas the observations are the geophysical variable to be predicted [20]. This process is repeated for each station location independently. The below three points summarize the main steps of the process at one FLT:

1. Starting from a current deterministic NWP forecast, the best matching historical forecasts (analogs) for the current prediction are chosen, at the same FLT (see Equation (2));
2. For each FLT, the observations corresponding to the analog forecasts are retrieved from the historical data set;
3. These observations form the analog ensemble future prediction at that location and FLT [20].

The AnEn technique adjusts the model bias by taking into account past errors, assuming that the model error can be estimated in finding a similar past forecast [37].

### 2.5. AnEn Generation

The AnEn has been generated using the PAnEn package, an integrated package for parallel ensemble forecasts, implemented in R and C [24].

The year 2015 has been used as the search past repository, while 2016 has been used as test time. The number of analog members has been fixed to 21. The optimal number of ensemble members is still an open debated issue; it depends on the size of the data repository and it could vary for each lead time. Initial studies show that a good number of ensemble members is the square root of the size of the historical search space [38,39]. The generated analogs are ranked based on their similarity (Equation (2)), from the most to the least similar.

The similarity metric describes the quality of the analog chosen and is based upon the similarity of the current deterministic forecast window to the past forecast time windows available in the historical repository. The similarity metric is computed using each of the predictors and can be thought of as a multivariate vector. Once the similar forecasts are chosen, the corresponding observations for each of the 21 analogs are selected. The corresponding observations, together, generate the 21 members of the ensemble of analogs, the prediction, for the current FLT.

A weight based on its prediction power is assigned for each of the 9 predictors, according to the findings of Clemente et al. [25], see (Table 1). The whole AnEn generation process is repeated independently for each PWS and each time step.

### 2.6. Spatial Downscaling

Although current applications of AnEn mostly involve generating forecasts at the same temporal and spatial resolution as the forecast model, we propose its application also in model downscaling.

The AnEn technique allows for spatial downscaling, by returning analogs for each of the PWSs. Only one NAM-NMM grid point is used to generate the analogs in this study, and 83 quality checked WU stations. The downscaling process of the NAM-NMM model, allows matching of the generated analogs to all the 83 WU stations.

Once the analogs for the NAM-NMM point are generated, the spatial downscaling process is executed by reusing the AnEn indices and assigning them to the corresponding time steps of each of the 83 observations' locations that are the target stations to which the analogs will be downscaled to. The downscaling is run for all the 365 test days and for all the FLTs, generating downscaled analog predictions results over the entire domain area, and over the entire test period (see Figure 5 for a summary of the main steps).

---

1) Assign the NAM-NMM grid point to each WU PWS, based on the eucliadian distance

⬇

2) For each test days retrieve the most matching historical forecasts (i.e.: day 1 = 1, 3, 75; day 2 = 4, 64, 107)

⬇

3) For each WU PWS retrieve the corresponding observed measurements using the AnEn indices (i.e.: day 1 = $x^1$, $x^3$, $x^{75}$; day 2 = $x^4$, $x^{64}$, $x^{107}$)

⬇

4) Repeat the process for all the 365 test days and for all the FLTS

---

**Figure 5.** Main steps of the proposed downscaling indices method. In bullet point three, $x^j$ indicates the historical observation, and *j* represents the historical forecast.

The downscaling approach presented here is a revised version of the statistical down-scaling used in the AnEn technique literature [14,40,41]. It is important to notice that only the AnEn technique allows matching of past model's forecasts and past observation measurements through their correspondent indices composed by the combination of days and FLTs. This is the first time this downscaling technique based on the use of indices is used in combination with the AnEn technique, and its innovative use to predict the temperature in an urban environment.

## 3. Results

The AnEn generated temperature forecasts spatially downscaled to the 83 WU stations over Manhattan.

Figure 6 shows the distribution of the downscaled temperature analogs for 8 August 2016, at 12 p.m. local time (correspondent to the 16th FLT), over the 84 h.

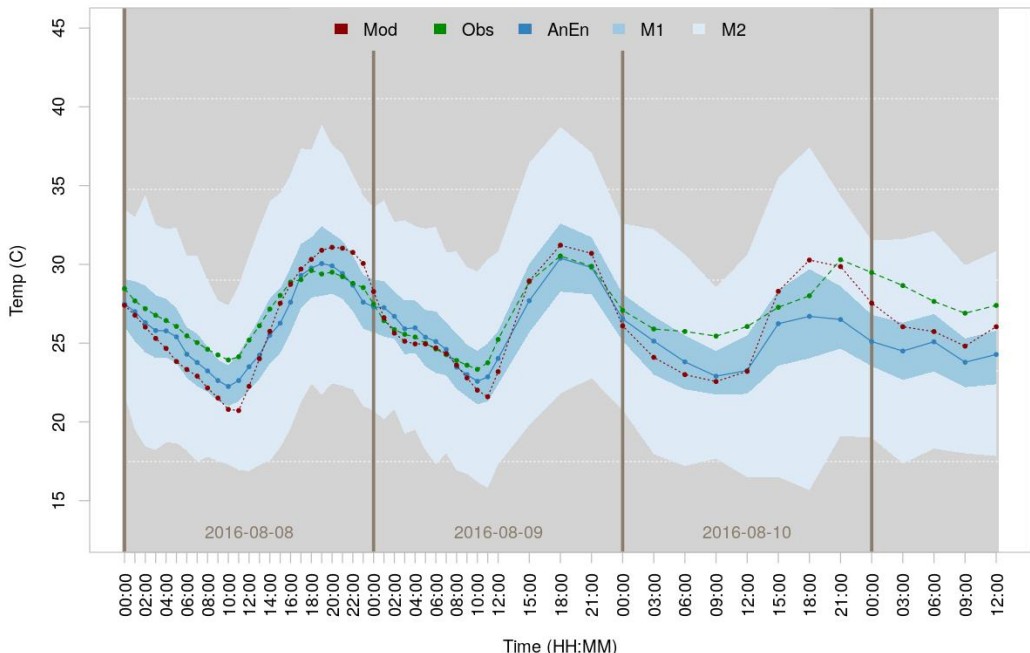

**Figure 6.** AnEn temporal representation of the downscaled forecast for 8 August 2016. The figure shows the original hourly NAM-NMM forecast (red dotted line), the average PWSs interpolated accordingly with the NAM-NMM model (green dotted), the AnEn ensemble mean (blue dotted), and the M1 and M2 points (shaded blues).

The dotted red line represents the NAM-NMM model forecast prediction over the 84 FLTs. The dotted green line represents the WU observation measurements, interpolated accordingly to the time steps of the model. The dotted blue line represents the AnEn ensemble mean—across stations and analog members. The upper (M2) and lower (M1) limits represent the third and first quartile (75th and 25th percentile) respectively for the distribution of the analogs. By default, M2 will extend up to 1.5 times the interquartile range from the top and bottom of the box to the furthest datum within that distance.

Figure 7 shows the results of the spatial downscaling, for the same day and time represented in Figure 6 (8 August 2016—12 p.m. local time). The upper left map shows the PWSs' observations interpolated over the domain area using the Inverse Distance Weighted (IDW) algorithm; the upper right map shows a continuous surface of the AnEn temperature prediction generated by interpolating the ensemble mean of each of the 83 PWSs, using the same IDW algorithm; the lower left map shows the NAM-NMM temperature forecast, which is uniform because only one grid point has been used; and the lower right map shows the difference between the PWSs observed measurements and the AnEn ensemble

mean. It is important to note that the 95% of the AnEn bias prediction errors, presented in the fourth panel ((d)) of Figure 7, range between -2 C and 5 C. Errors can be attributed to the coarse resolution of the NAM-NMM model used to generate weather analogs, and this can make the prediction complicated for some PWSs. Moreover, accuracy can be affected by the unknown location of the PWSs, which can affect the quality of the analog predictions.

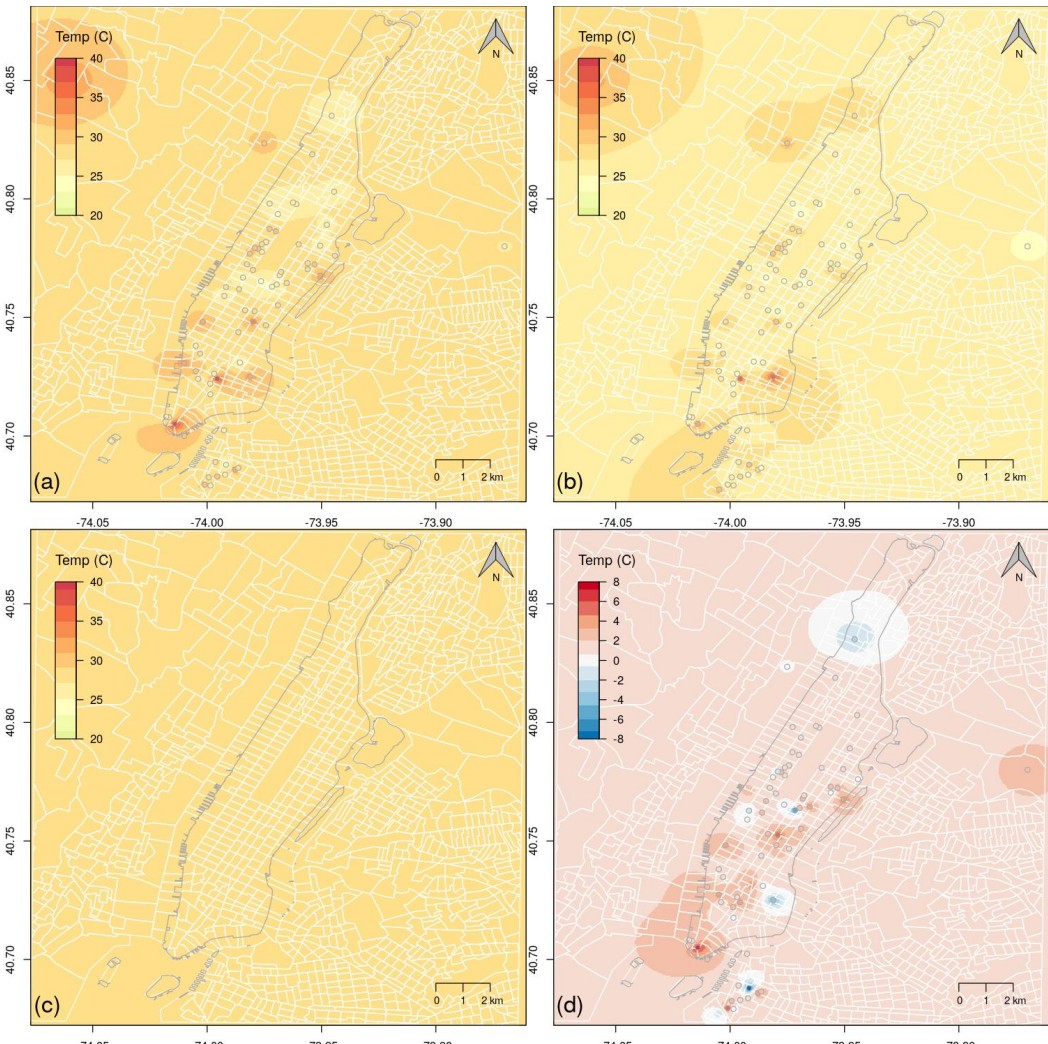

**Figure 7.** AnEn spatial downscaling for 8 August 2016 and forecast lead time 16 (12 p.m. local time)—The figure shows the observed values from the WU stations (**a**), the AnEn ensemble mean (**b**), the NAM-NMM prediction (**c**) and the difference between the PWSs and the AnEn ensemble mean (**d**).

Figure 8 shows the downscaled predictions' errors of the model and of the AnEn. More specifically, it shows the spatially interpolated RMSE, of the NAM-NMM and of the AnEn as a function of lead times; the map on the left shows the RMSE averaged over each of the 365 test days, and interpolated using IDW, between PWSs and NAM-NMM. Similarly, the map on the right shows the RMSE between AnEn and the 83 PWSs, calculated over the entire test set and interpolated using IDW. The overall mean error of the AnEn (2.759 °C) is smaller than the overall mean error of the NAM-NMM model (3.208 °C).

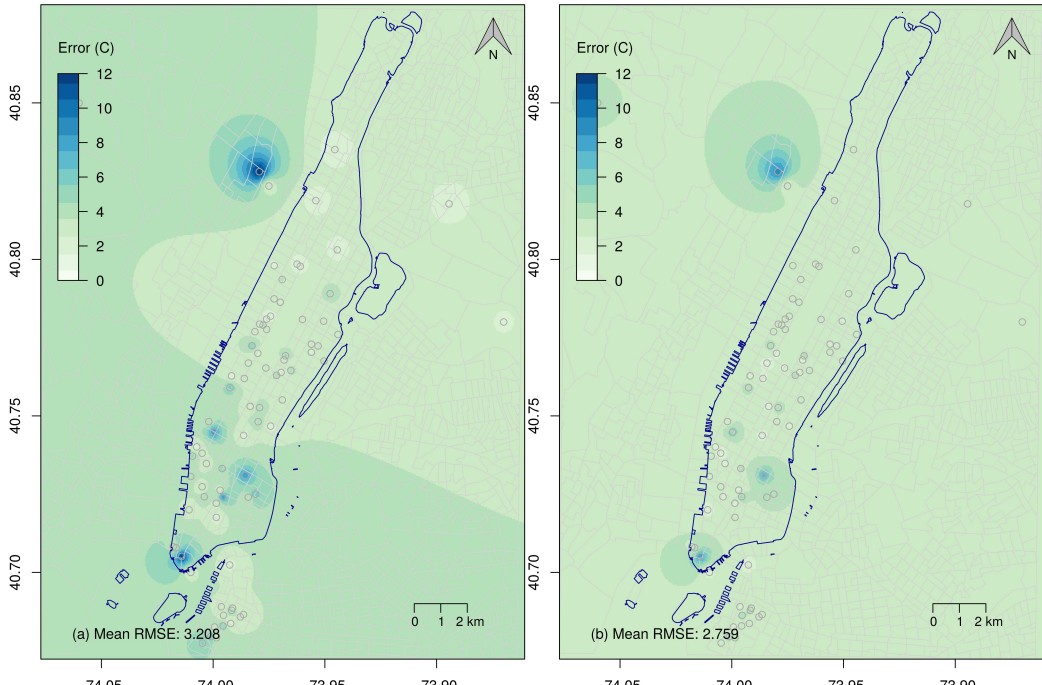

**Figure 8.** Downscaled prediction error map—The figure shows the spatial representation of the RMSE of NAM-NMM and AnEn as a function of lead times; the (**a**) map on the left shows the mean spatial error calculated over the entire test period between PWSs and NAM-NMM, while the (**b**) map on the right shows the mean spatial error between PWSs and the AnEn.

Figures 6 and 7 show the ability of the AnEn technique and of the downscaling approach here presented, to accurately predict temperature even if only one NAM-NMM model grid point is used as input. Furthermore, the AnEn results, for each day and FLT, can be represented as a continuous, non-flat, surface, which shows spatial differences within the domain. The main advantage of the downscaled AnEn is indeed represented by the added value given by the spatial distribution of the values within the domain.

The quality of the AnEn forecasts—the degree to which they correspond to what has been measured by the PWSs, is assessed using different statistical measures that return information about the forecast errors, quantifying consistency, reliability, resolution and skill of the forecasts.

The verification measurements are usually performed by comparing the forecasts to their corresponding observations. Figure 9 shows the distribution, over the FLTs, of four different verification measurements for both the four stations, the other 79 PWSs—the blue step-lines refer to the AnEn predictions, while the red step-lines refer to the NAM-NMM model forecasts, the NAM-NMM model and for a Persistence (Pers) prediction.

NAM-NMM generates deterministic forecasts, and thus to compare ensemble means, a Pers forecast is generated as a benchmark starting from the NAM model. The Pers forecast is often used as a standard of comparison between different forecast methods, especially when the prediction refers to a very short projection [42,43]. The Pers model generates an ensemble by taking the previous *n* forecasts. In this case, the value *n* was set to 21 to correspond to the number of ensemble members used to generate the analogs.

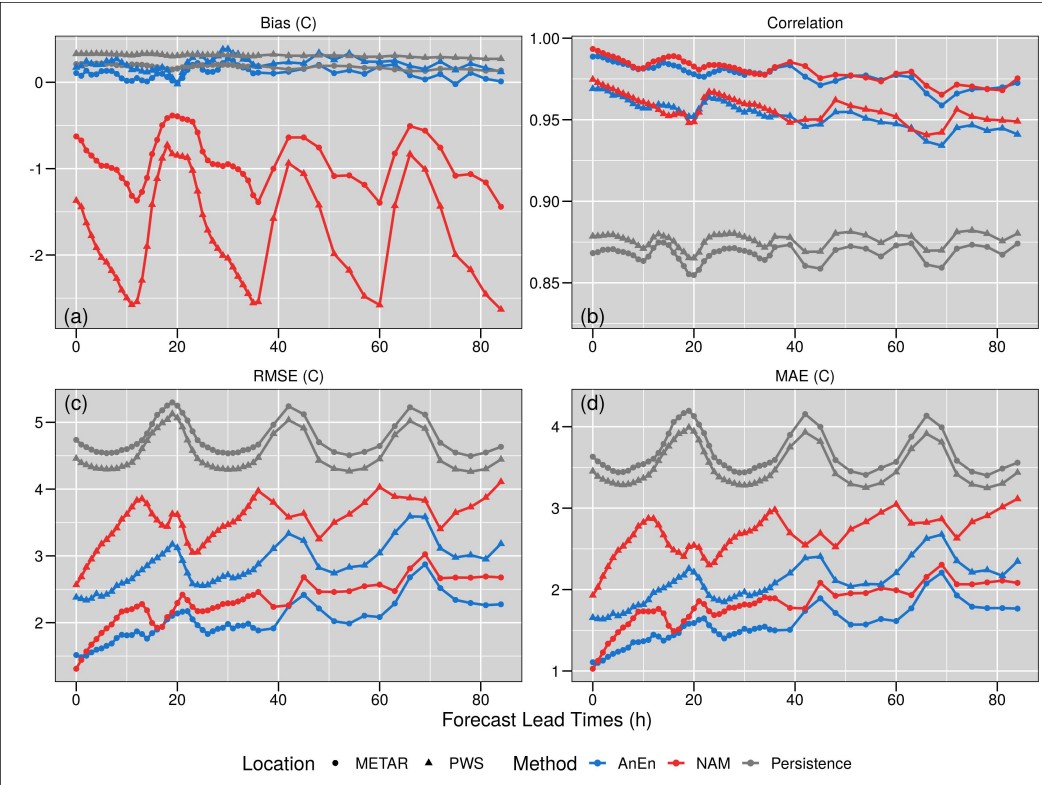

**Figure 9.** Verification measurements—The figure shows the verification measurements used to verify the performance of the AnEn prediction of the four stations and the other PWSs (blue lines) in comparison to the Pers (dark grey lines) and to the downscaled NAM-NMM forecasts model, in terms of bias (**a**), correlation (**b**), RMSE (**c**) and Mean Absolute Error (MAE) (**d**).

The (a) panel shows the bias, the correspondence between the mean forecast and mean observation. The best assumption would be a bias value of 0. The four PWSs show an AnEn average bias value equal 0.11, while the other 79 PWSs show an average bias value equal to 0.22 °C; these values are compared with the downscaled NAM-NMM forecast model that shows for the official ones an average bias of −0.91 °C, and for the other stations a value of −1.79 °C. The Pers shows a bias of 0.183 °C for the stations, and a bias of 0.312 °C for the other PWSs. The graph shows how both the PWS and the METARs stations perform better, in terms of bias, if compared to the NAM-NMM model, and slightly better if compared to the Pers.

The (b) panel shows the correlation, which indicates a measure of the linear association between forecasts and observations independent of the mean and variance of the marginal distributions. A perfect correlation score corresponds to 1. The AnEn for the four PWSs shows an average correlation value equal to 0.98 °C, while the other 79 PWSs shows an average bias value equal to 0.96 °C; the downscaled NAM-NMM model shows for the official ones, an average correlation of 0.98 °C, and for the others a value of 0.96 °C. The Pers shows a 0.868 °C correlation for the METARs stations, and a 0.876 °C correlation value for the other PWSs. Looking at the correlation values and graph, AnEn performs better if compared to the Pers, but shows similar performance if compared to the NAM-NMM model.

The (c) panel shows the RMSE. It represents the square root of the average of the squared differences between forecasts and observations. Perfect score equals to 0. The AnEn for the METARs stations shows an average RMSE value equal to 2 °C, while for the other 79 PWS stations reports an average RMSE value equal to 2.82 °C; the downscaled NAM-NMM model shows for the METARs stations, an average RMSE of 2.28 °C, and for the others a value of 3.52 °C. The Pers shows a value of 4.758 °C for the METARs stations, while for the other a values of 4.532 °C. In terms of RMSE, the downscaled AnEn

performs better if compared to the Pers forecasts and slightly better if compared to the NAM-NMM model.

The (d) panel shows the MAE. It is an error that is represented by the average of the absolute differences between forecasts and observations. The perfect score is 0. The downscaled AnEn for the METARs stations shows an average value equal to 1.52 °C, while for the other PWS reports an average value equal to 2.02 °C; the NAM-NMM model shows for the METARs stations an average MAE of 1.76 °C, and a value of 2.65 °C for the other PWSs. The Pers reports a MAE of 3.665 °C for the METARs stations and a 3.494 °C value for the all the other PWSs. Additionally in terms of MAE, the AnEn performs better than the Pers and the NAM-NMM model, showing a worse performance that increase over the prediction time.

By generating the verification measurements presented, we also calculated the 95% confidence interval of the mean difference by performing a bootstrap resampling. Figure 9 shows the mean values for each measurement, while Table 2 presents the upper and lower bounds, for two days forecasts, for both the four METARs stations and the PWSs.

**Table 2.** Bootstrap confidence intervals of the verification measurements.

| Station Type | Metric | 95% CI | Lead Time (h) | | | | | | | | |
|---|---|---|---|---|---|---|---|---|---|---|---|
| | | | 0 | 6 | 12 | 18 | 24 | 30 | 36 | 42 | 48 |
| Official METARs stations | Bias | Lower | 0.023 | 0.05 | −0.048 | −0.043 | 0.123 | 0.167 | 0.016 | 0.012 | 0.086 |
| | | Upper | 0.186 | 0.224 | 0.142 | 0.173 | 0.324 | 0.374 | 0.207 | 0.234 | 0.315 |
| | Correlation | Lower | 0.987 | 0.983 | 0.98 | 0.978 | 0.977 | 0.975 | 0.98 | 0.974 | 0.971 |
| | | Upper | 0.99 | 0.986 | 0.983 | 0.983 | 0.981 | 0.98 | 0.983 | 0.979 | 0.976 |
| | RMSE | Lower | 1.413 | 1.577 | 1.77 | 1.935 | 1.863 | 1.873 | 1.792 | 2.146 | 2.108 |
| | | Upper | 1.598 | 1.722 | 1.953 | 2.151 | 2.052 | 2.071 | 1.959 | 2.338 | 2.311 |
| | MAE | Lower | 1.052 | 1.205 | 1.379 | 1.482 | 1.43 | 1.452 | 1.434 | 1.672 | 1.641 |
| | | Upper | 1.161 | 1.314 | 1.506 | 1.623 | 1.568 | 1.586 | 1.562 | 1.815 | 1.786 |
| PWS stations | Bias | Lower | 0.139 | 0.21 | 0.116 | 0.103 | 0.292 | 0.345 | 0.142 | 0.184 | 0.304 |
| | | Upper | 0.205 | 0.279 | 0.19 | 0.185 | 0.363 | 0.423 | 0.221 | 0.276 | 0.376 |
| | Correlation | Lower | 0.968 | 0.962 | 0.957 | 0.952 | 0.962 | 0.953 | 0.951 | 0.944 | 0.953 |
| | | Upper | 0.97 | 0.965 | 0.96 | 0.956 | 0.964 | 0.956 | 0.954 | 0.948 | 0.956 |
| | RMSE | Lower | 2.334 | 2.374 | 2.683 | 3.02 | 2.524 | 2.668 | 2.822 | 3.272 | 2.782 |
| | | Upper | 2.418 | 2.467 | 2.774 | 3.152 | 2.6 | 2.758 | 2.924 | 3.394 | 2.862 |
| | MAE | Lower | 1.634 | 1.681 | 1.936 | 2.148 | 1.857 | 1.943 | 2.053 | 2.351 | 2.085 |
| | | Upper | 1.68 | 1.729 | 1.99 | 2.208 | 1.906 | 1.992 | 2.106 | 2.419 | 2.135 |

This study focuses on deterministic prediction, but briefly discuss the distribution of the ensemble would clear the potentiality of the downscaling method proposed for future research works that can be used to generate downscaled probabilistic prediction, using the AnEn technique.

To assess the overall quality of the ensemble prediction, and to evaluate the performance of the probabilistic prediction, we generated the Continuous Ranked Probability Score (CRPS) verification measurement, by comparing the integrated squared difference between the Cumulative Distribution Function (CDF) of the forecasts and the corresponding CDF of the observations [37,44]. The general rule to interpret the CRPS output, says the lower the value the better the performance, with the perfect score is equal to 0. The CRPS is defined in Equation (3), as follows:

$$CRPS = \frac{1}{N} \sum_{i=1}^{N} \int_{-\infty}^{\infty} \left( F_i^f(x) - F_i^0(x) \right)^2 dx \qquad (3)$$

where $F_i^f(x)$ is the CDF of the probabilistic forecast and $F_i^0(x)$ is the CDF of the observation for the $i$th ensemble observation/forecast combinations, and $N$ is the number of available combinations.

Figure 10 shows the CRPS generated for both the AnEn (blue line) and the Pers predictions (grey line, as a function of FLTs. The AnEn CRPS slightly worsen over time, while the one generated for the Pers seems to be more stable.

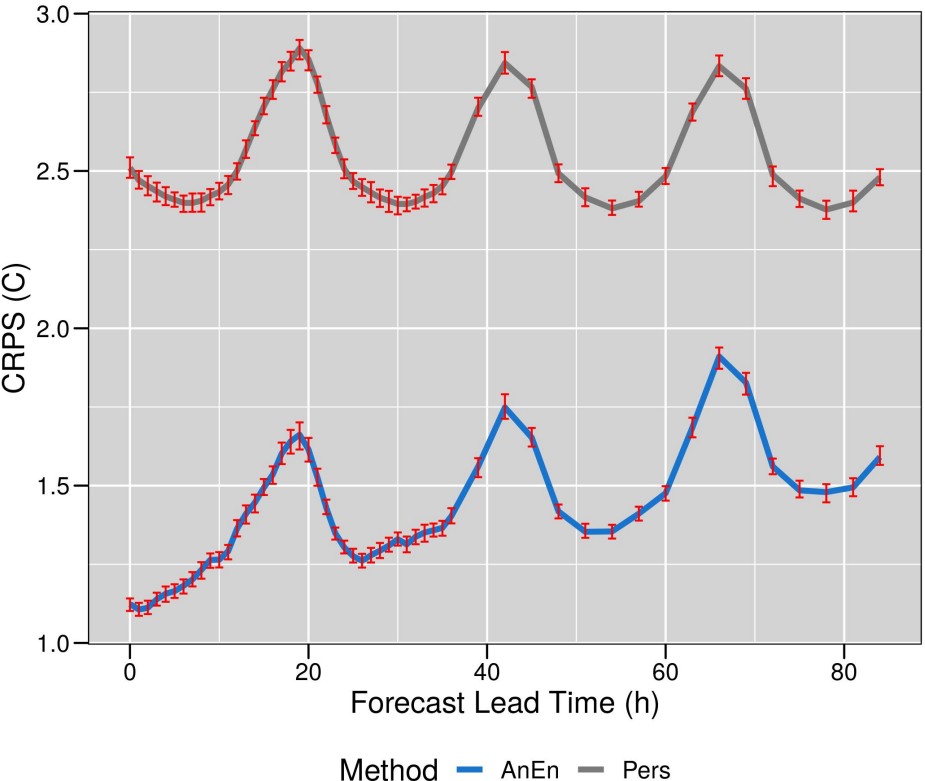

**Figure 10.** The figure shows the CRPS verification measurement used to evaluate the performance, as a function of FLTs, of the AnEn prediction (blue line) in comparison to the Pers (grey line). The error bars indicate the 95% bootstrap confidence intervals.

The red error bars represent the 95% confidence interval for the mean difference calculated by performing a bootstrap resampling. The mean CRPS value for AnEn is equal to 1.41 °C, while for the Pers the value equals to 2.54 °C.

These results show how the downscaled AnEn technique performs better in forecasting downscaled temperature in an urban environment if compared to a Pers forecast, and to the NAM-NMM model, in most of the cases, both when generating a deterministic and a probabilistic prediction.

## 4. Discussion and Conclusions

According to NOAA [45] the global average increase rate of combined land and ocean temperatures has more than doubled since 1981 to 0.18 °C a year, while for New York State this increase was equal to 1.5 °C (https://www.nytimes.com/interactive/2021/05/12/climate/climate-change-weather-noaa.html (accessed on June 2021)).

However, warming is not uniform across the planet [46]. The impacts of climate change can be seen across scales, and they can be amplified by local conditions, or mitigated by global circulation. Therefore, certain areas of the planet, including many megacities, can have more dangerous warming trends than what is expected globally [47]. Stone [48] proposed a theory where our society is seen as an urban planet. Under this assumption, the major impacts of climate change are expected to take place in cities. This implies that the generation of accurate and high-resolution forecasts in an urban environment is a primary defense to mitigate the effects of climate change and help urban population to cope with the increasing exposure to high temperatures.

This work shows how it is possible to use contributed data, which are predicted to become always more abundant, to downscale weather model forecasts and provide high-resolution forecasts in densely populated areas. High-resolution downscaled forecasts are

paramount to protect people, properties and the environment from increasing temperature trends and periods of prolonged extreme heat.

The AnEn technique is used to generate downscaled ensemble forecasts using initial deterministic forecasts from the NAM-NMM weather prediction model, and a set of historical observations from the WU network. The original NAM-NMM deterministic forecasts are downscaled to a much finer spatial resolution obtained by interpolating the measurements of the dense PWSs network. The results show that it is possible to use contributed and non-quality-controlled WU observations to improve the original forecasts. Experiments were performed using two years of data for the Manhattan borough.

A quality control for the WU data was performed by comparing each of the 105 stations with four PWSs which were present in the dataset. A method was developed to filter out entire stations or portions of the data when the observations greatly deviated from the mean of the four control stations.

The forecasts from AnEn were compared to the original forecasts of the NAM-NMM model, and to a simple persistent analog model Pers. This latter model was generated to determine the improvement of the forecast when compared to a very simple model.

The AnEn deterministic predictions perform better than the Pers and the NAM-NMM model, showing a better accuracy, and CRPS shows a better ensemble quality when compared with Pers. Results show the capability of the AnEn to generate accurate and calibrated forecasts that are spatially distributed at each of the WU stations over the

The higher resolution of the downscaled AnEn predictions can identify specific areas where high temperature can potentially affect population's health. The ability of downscaling original forecasts can drastically improve health and risk management, and be particularly actionable during heat waves. This feature is particularly important given the current weather trends of more dramatic and frequent extreme temperatures, paired with a growing urban development, population density and social disparity. Accurate and high-resolution forecasts are paramount to help improving the resilience of an urban population. The AnEn has not previously been combined with VGI data to generate temperature forecasts and to downscale the predictions at a higher spatial resolution. The results inform that it is possible and desirable to use datasets that differ in spatial resolutions.

Future work will demonstrate how PWSs data and the AnEn technique can be paired for predicting extreme weather events, and to extend the downscaling to the temporal dimension.

**Author Contributions:** M.C., W.H. and G.C. contributed to all aspects of the work: background review, data development, analyses, writing. L.D.M. contributed to background review and writing. All the authors contributed to project planning. All authors have read and agreed to the published version of the manuscript.

**Funding:** This research was funded by the Office of Naval Research (ONR) award #N00014-16-1-2543.

**Data Availability Statement:** The NAM-NMM forecasts data were derived from the NCEP—National Oceanic and Atmospheric Agency (NOAA) website resources available in the public domain: https://doi.org/10.5065/G4RC-1N91 (accessed on March 2020). The WU observation measurements data were downloaded from the WU website, upon an API request at the https://www.wunderground.com/ (accessed on October 2016).

**Acknowledgments:** We wish to thank Olga Wilhelmi (Research Applications Laboratory, National Center for Atmospheric Research) and Francesca Chiaromonte (Huck Institutes of the Life Sciences, The Pennsylvania State University) for their comments that helped improve the present manuscript.

**Conflicts of Interest:** The authors declare no conflict of interest.

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
