# Peer review of "NAM-NMM Temperature Downscaling Using Personal Weather Stations to Study Urban Heat Hazards"

_2624-795X, doi:10.3390/geohazards2030014_

Round 1

Reviewer 1 Report

Manuscript Review

Title: WRF-NAM Temperature Downscaling using Personal Weather Stations to Study Urban Temperature Heat Hazards

Review:  The manuscript describes the use of the Analog Ensemble technique to statistically downscale temperature forecasts generated by the NAN-NMM numerical weather prediction model.  The goal of the study is to produce high-resolution forecasts for the borough of Manhattan, NY that could be used to assess the effects of local effects of urban heat islands on near-surface temperature. High-resolution climate and weather information is vital for the study of climate change effects at regional and local scales, and the development of accurate statistical and dynamical downscaling methods is crucial for the generation of such information.

Studies have shown that downscaling can add significant value to coarse-resolution climate and weather model forecasts and the implementation of new and reliable high-resolution downscaling techniques is needed. Despite some very interesting results, the manuscript falls short of delivering an organized and clear description of the steps leading to them. Critical information about the methodology is missing or confusing. For instance, the authors state that “AnEn forecasts are intrinsically calibrated and bias corrected for the specific variable being predicted but I there is no evidence of that in the text. Additionally, a few statements are not very clear, figures are missing captions, and values are missing units.  Below are my major review comments and recommendations as well as minor corrections suggestions, which I believe would improve the manuscript’s scientific soundness.

Recommendation: accept only after major revisions

Major review comments:

  • The title indicates that WRF-NAM model results are used for the downscaling, but section 2 describes the North American Mesoscale Forecast System (NAM-NMM) model is used. Are these models the same?  If so, please choose one name and explain that the abbreviation means (WRF: weather and research forecasting model, NMM: Nonhydrostatic Mesoscale Model).  Maybe you are correct model name is WRF-NMM
  • Line 144: Please explain how the 5 predictors were selected out of the 9 available.
  • Figure 1: Legend text is repetitive.  Remove the statement: “The green points represent the included stations, while the reds the excluded stations, meaning these have a mean annual temperature that deviates more than 3 °C from the threshold”.  It is clear from the first part of the legend that the red points are excluded.
  • Line 216: This statement is not clear “The NAM-NMM stations are generated on a grid points”.  I believe that you meant to say that the model results are distributed on a grid.  Please describe the NAM-NMM grid.  Model grid information is critical to assess the quality and added value of the downscaling product
  • Line 224: Forgot the add the RMSE unit.  Not only here but throughout the paper, units are missing
  • Line 215: NAM-NMM station identification.  It is not clear why one 1 of the model grid points was used for the downscaling.  Despite having slightly higher RMSE, wouldn’t the use of all 3 stations improve the downscaling overall results?  Please explain.
  • Line 232: please explain how the AnEn forecasts are intrinsically calibrated and bias corrected for the specific variable being predicted. 
  • Line 238: Please explain what you mean by an “invariant model”
  • Line 240: What type of relationship has to be constant?   Model and observation grid?  Please explain
  • Line 348-349: Please add a reference to this statement and explain how this is relevant to your study
  • Figure 6, 7, 8:  Please add A, B, C captions in the figure and include these in the legend to describe the different panels. 
  • Line 397: please add a reference for the Pers forecast mentioned in the manuscript. 
  • Line 401: Please confirm that the citation #6 a reference for this statement: A Pers forecast generally assumes “the tendency for the occurrence of a specific event to be more probable, at a given time, if that same event has occurred in the immediately preceding time period“6
  • Lines 407-409: units missing
  • Line 418:  What is the statistical significance of these correlations?  P-values?
  • Line 433-434: Is the 0.18°C estimate a global average?  What is it for your study area?

Minor editing:

Line 171: number of stations

Line 273: has been proven/proved

Line 330: A weight based on its prediction power is assigned for each of the 9 predictors

Line 332: respectively

Reviewer 2 Report

General comments:

Overall, I found this paper interesting, but needs significant revising to make it suitable for publication.  I find the use of PWS data, through quality control, interesting and unique.  The application of AnEn is less unique but still interesting.  I would also change title to NAM-NMM (as used in text) instead of WRF-NAM.

The main issues I have with paper are:

  • Introduction and discussion discuss impact of UHI on future climate, but this is not addressed in this paper. This paper focusses on short term forecasts.  Sure, there might be implications for downscaling climate data, but it is not clear how this study will impact downscaling data.
  • Discussion of processing of PWS data is not written clearly. It is not clear the final process to decide which PWS data used. Initially, some stations are removed because annual bias relative to the METAR stations is greater than 3o C, but then they make further decisions on an hourly basis. This needs to be made clearer.
  • Statistical significance of results is not discussed. Also, how much improvement is due to bias correction vs using the AnEn approach?
  • The discussion is focussed on the ensemble mean performance of AnEn approach. Maybe also discuss if there is any useful information related to the distribution of the ensemble?

Detailed comments:

Pg2/ln 38: Assume you mean weather forecast models instead of Atmospheric? Also, it appears her that you are now going to focus more on short term forecasts vs climate projections. This should be stated.

Pg2/ln 47-49: I would state that the likely main failure of model forecasts is related to resolution itself and the specification of the land surface type.

Pg2/ln 66-67: Not just convective cooling, also radiative effects are important.

Pg2/ln 73: note climate models only give projections, not forecasts. Note that for projections, there is likely greater uncertainty related to uncertainties of the climate projections than due to local effects.  This needs to be stated.

Pg3/ln 82: Unclear what is meant by ‘concurrent interventions’. I assume you mean boundary forcing 9by whatever means the downscaling is completed).

Pg3/ln87-88: Not sure what is meant by sentence stating with “Furthermore…”. Due to numerical constraints, dynamical downscaling typically uses finer timesteps. Also, temporal frequency is more controlled by storage requirements than model resolution.

Pg3/ln 113-4: Not clear what is meant by “allows errors to be averaged out” as differences between stations is what is wanted?  Errors are not averaged out, differences are.

Pg4/ln 141:  What is meant by “intercurs”? Maybe remove “that intercurs” and add “forecast initial time

Pg4/ln 145 and Table 1: Interesting weight of .7 for soil temp, but only .1 for air temp. Need discussion on how/why these weights were determined.

Pg4/ln 155:  I assume you mean “for the purpose of this study” instead of “to the study purpose”. Also, remove “accordingly”.  Also, does this mean the higher frequency data (between hours) is not used?  The higher time frequency of the PWS cwas mentioned earlier, assuming this meant it was useful?

Pg4/ln 157:  As mentioned above, this section needs work to make clearer the final process of quality controlling which station data was used.

Pg5/ln 180: State there are 105 stations under consideration here, but later on, authors talk of only 100 station considered (ln 213). What happened to other 5 stations?

Pg5/ln 191: Add ‘hourly’ before “mean temperature” to be clear.

Pg6/ln 198-200: Is this just a repeat of what was said earlier?

Pg6/Figure 2: X-axis is mislabelled as these are not lead time since these are observations? Not clear what is X-axis; hours relative to what. How is data averaged.

Pg7/ln 210: Rephrase “systematic polishing phase” to something like “refinement”

Pg7/ln 213: The 17 stations excluded were from the 100 left after the 5 excluded station which had no data?

Pg7/ln 214: Add at end of line “at least for some times”. Question: Has does including one time affect time window?

Pg9/ eqn 1: Need to specify what xf is (and what superscript f means).

Pg9/ln 273: Change proof to proved.

Pg10/ln 292-3: Do not se that AnEn provides “generation in real-time of a higher fidelity prediction (with finer horizontal and vertical resolution, higher-order numerics, more advanced physical schemes, etc.)” Finer resolution, yes (due to denser PWS network), but not the others since AnEn is purely statistical.

Pg10/ln 305: How is “best match” decided.  Need more details here.

Pg10/ln 332-3: Why are these weights different than in Table 1?

Pg12/ Figure 6: Why do stations still show warm/cool bias?  Is this because AnEn is picking historical analogue values which are hotter/cooler than for this actual day on average?  Needs some discussion. Also, would prefer a more discrete colour scale to more easily determine values. Maybe pick smaller range would help as well. Need to label subplots and explain in caption what they are.

Pg14/Figure 8: Why does persistence have non-zero bias?

Round 2

Reviewer 1 Report

Manuscript Review

Title: NAM-NMM Temperature Downscaling using Personal Weather Stations to Study Urban Temperature Heat Hazards

Review:  The revised version of the manuscript is a substantial improvement compared to the original. The authors have addressed all of the issues raised by me during the first round of reviews, including the lack of clarity in the methodology, missing figure captions, missing units, and some critical information about the statistical model and data interpretation used for the study. Overall, the authors have provided satisfactory answers to all concerns and updated for manuscript accordingly.  Additionally, they managed to incorporate some of my suggestions and correct the typos found in the original submission. I would like to congratulate the authors for the nice work.

I recommend that this revised version of the manuscript be accepted for publication.

Recommendation: Accept

This manuscript is a resubmission of an earlier submission. The following is a list of the peer review reports and author responses from that submission.